# Fast and Robust Early-Exiting Framework for Autoregressive Language Models with Synchronized Parallel Decoding

**Sangmin Bae**[1*]   **Jongwoo Ko**[1*]   **Hwanjun Song**[2†]   **Se-Young Yun**[1†]

[1]KAIST AI    [2]AWS AI

{bsmn0223, jongwoo.ko, yunseyoung}@kaist.ac.kr    hwanjuns@amazon.com

https://github.com/raymin0223/fast_robust_early_exit

## Abstract

To tackle the high inference latency exhibited by autoregressive language models, previous studies have proposed an early-exiting framework that allocates adaptive computation paths for each token based on the complexity of generating the subsequent token. However, we observed several shortcomings, including performance degradation caused by a state copying mechanism or numerous exit paths, and sensitivity to exit confidence thresholds. Consequently, we propose a Fast and Robust Early-Exiting (FREE) framework, which incorporates a shallow-deep module and a synchronized parallel decoding. Our framework enables faster inference by synchronizing the decoding process of the current token with previously stacked early-exited tokens. Furthermore, as parallel decoding allows us to observe predictions from both shallow and deep models, we present a novel adaptive threshold estimator that exploits a Beta mixture model to determine suitable confidence thresholds. We empirically demonstrated the superiority of our proposed framework on extensive generation tasks.

## 1 Introduction

Recent advancements in autoregressive language models (Brown et al., 2020; Raffel et al., 2020; Hoffmann et al., 2022; Touvron et al., 2023) have revolutionized the quality of language generation in various generative tasks, including question answering (Rajpurkar et al., 2016a), summarization (Nallapati et al., 2016; Fabbri et al., 2019b), and machine translation (Cettolo et al., 2017a). Nevertheless, these large transformer models have shown high inference latency due to the considerable number of layers and the autoregressive decoding step. As the multiple stacks of transformer layers have to be computed sequentially for each individual token, the inference process poses significant computational burdens and hinders their real-time adaptability (Jiao et al., 2020).

In light of the necessity to expedite inference latency, the *early-exiting* framework (Elbayad et al., 2020; Liu et al., 2021; Schuster et al., 2022) emerges as a promising approach that dynamically allocates computation paths based on the complexity of generation for each token. As illustrated in Figure 1a, tokens that are relatively easy to predict the subsequent token yield consistent predictions with only a few layer computations, while those with higher difficulty require computations across a larger number of layers to generate accurate predictions. In an ideal scenario, the early-exiting method empowers models to achieve notable acceleration in inference without compromising the generation quality when compared to that of a full model.

However, our extensive analysis identified four challenges in the early-exiting framework. *Firstly*, despite the potential to exit at earlier layers, key and value states for remaining layers are still required for processing subsequent tokens. While previous works have proposed the state copying mechanism (Elbayad et al., 2020; Schuster et al., 2022) to efficiently compute these states by reusing hidden states from the early-exited layer, our findings reveal that this method performs poorly with larger models and longer output sequences (see Section 4.1). *Additionally*, setting all layers as possible exit positions does not guarantee faster inference due to (1) the defective performance of earlier layers that can generate abnormally long sequence outputs, and (2) the computational overhead from confidence measurement at every layer (see Section 4.2 and 4.3). *Lastly*, achieving the desired level of latency and accuracy with early-exiting heavily depends on selecting the appropriate confidence threshold for the target task. This often entails significant efforts and additional computational overhead (see Section 4.4). Hence, these challenges call for a new approach that consistently

---

*equal contribution    †corresponding authors

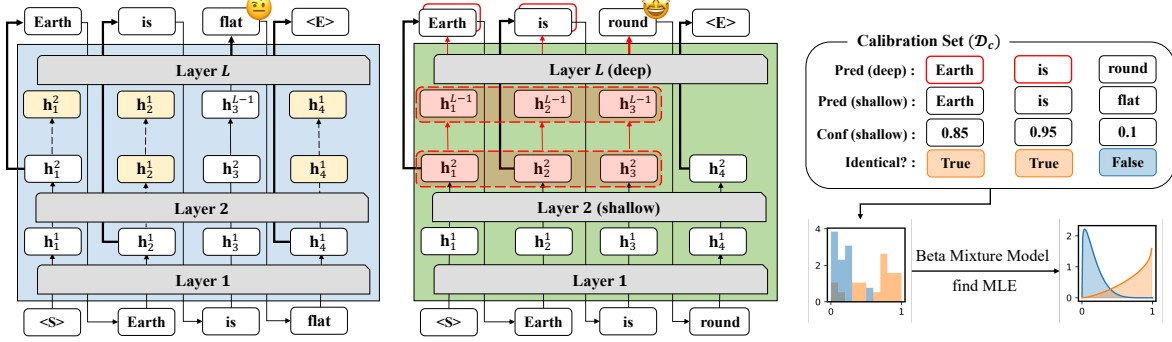

(a) Conventional Early-Exiting

(b) Fast and Robust Early-Exiting (FREE)

Figure 1: Overview of our FREE framework compared to the conventional early-exiting framework. FREE exhibits three key differences: (1) FREE employs a shallow-deep module that utilizes two exit points instead of employing all layers as exit points, (2) FREE replaces the state copying mechanism (yellow colored) with synchronized parallel decoding (red colored) to prevent performance degradation while accelerating inference speed, and (3) FREE utilizes an adaptive threshold estimator to determine the appropriate threshold values for each dataset during inference.

demonstrates high performance and low latency across diverse language models and datasets.

In this paper, we introduce a **F**ast and **R**obust **E**arly-**E**xiting (**FREE**) framework that incorporates a shallow-deep module and synchronized parallel decoding. Our framework not only offers consistent speedup and performance even for larger models and longer output sequences, but also eliminates the need for the computationally expensive process of finding the appropriate exit threshold.

Specifically, the *shallow-deep* module bifurcates the computation paths into a shallow model (with a specified number of early layers) and a deep model (including all layers). Our *synchronized parallel decoding* accumulates consecutive early-exited tokens that only pass through the shallow model until a non-exiting token is encountered. Thereby, we synchronize the decoding process of the current non-exiting token with the previously stacked tokens, as shown in the left of Figure 1b. This prevents performance degradation by utilizing actual attention computed key and value instead of approximated states through state copying, while it also achieves a more efficient approach compared to decoding each token autoregressively. Furthermore, we devise a novel *adaptive threshold estimator*, as shown in the right of Figure 1b, by leveraging the fact that parallel decoding outputs predictions even for early-exited tokens from the deep model. This estimator uses a Beta mixture model (BMM) to capture the correlation between confidence scores and prediction alignment of two models, determining the proper confidence threshold for each dataset. In practice, we demonstrate the efficiency of our FREE framework on extensive generation tasks.

## 2 Related Work

### 2.1 Early-exiting Framework

As the size of language models has significantly increased, there have been numerous efforts to develop efficient decoding methods that reduce the computational cost of language generation tasks. Motivated by prior literature (Teerapittayanon et al., 2016; Graves, 2016; Zhang et al., 2019a), Elbayad et al. (2020) introduced an early-exiting framework for faster inference, which dynamically adjusts the depth of the decoder for each token generation by making predictions at an intermediate layer. To achieve the better trade-off between speed and accuracy, Schuster et al. (2022) recently explored confidence thresholding methodologies, including various confidence measures, a decaying threshold function, and a calibration method.

However, their experiments were primarily conducted on small-sized decoder models, necessitating further validation on larger models. In addition, their approaches require additional training time for statistical tests on the extra calibration sets, which prevents them from real deployment scenarios.

### 2.2 Parallel Decoding

The non-autoregressive decoding, which generates multiple output tokens in parallel, was initially proposed by Gu et al. (2018). Several works (Ghazvininejad et al., 2019; Gu and Kong, 2021; Savinov et al., 2022; Santilli et al., 2023) have since focused on enhancing generation quality in machine translation tasks. Subsequently, Leviathan et al. (2023) introduced speculative decoding for sequence generation tasks. In this approach, an

approximation model (small size) predicts outputs autoregressively, while a target model (large size) runs in parallel to verify the acceptance of predictions made by the approximation model. With only accepted tokens, they resample the next token from an adjusted distribution. Related approaches have been proposed by Chen et al. (2023) and Kim et al. (2023), where they also utilize two models of varying depth and focus on refining the small model through speculative sampling or a rollback policy in a non-autoregressive manner.

Our approach is notably distinct from the aforementioned works as we focus on early-exiting framework by introducing synchronized parallel decoding within a *single* network, which incorporates a shallow-deep module. While we also leverage the advantage of simultaneously obtaining predictions from models of different depth, we rather aim to develop a novel and effective estimation methodology to adaptively determine the optimal threshold for each dataset. It is worth noting that their refining strategies may result in unbounded latency increase as they restart from incorrect predictions.

## 3  Preliminary

A Transformer network (Vaswani et al., 2017) is composed of $L$ layers, where each layer consists of two sublayers, a multi-head attention (MHA) layer and a feed-forward network (FFN) layer. The computation for hidden states at time step $t+1$ via stacked Transformer blocks is as follows:

$$\mathbf{h}_{t+1}^{\ell} = \text{Transformer}_{\ell}(\mathbf{h}_{t+1}^{\ell-1}), \ \ell \in [1, L],$$

where $\mathbf{h}_{t+1}^{0}$ is the embedding layer outputs of $y_t$ that represents the generated token at time step $t$.

After $L^{\text{th}}$ layer of the decoder network, the predicted token $\hat{y}_{t+1}$, is determined by the probability output from a softmax classifier $\mathbf{W}_L$:

$$p(y_{t+1}|\mathbf{h}_{t+1}^{L}) = \text{softmax}(\mathbf{W}_L^{\top}\mathbf{h}_{t+1}^{L})$$

However, unlike the standard LMs, the early-exiting framework enables the generation of a subsequent token in earlier layers by using $p(y_{t+1}|\mathbf{h}_{t+1}^{\ell})$. If the confidence score $c^{\ell}$ is larger than the predefined threshold, we can make a prediction at time step $t+1$ as $\arg\max p(y_{t+1}|\mathbf{h}_{t+1}^{\ell})$. While classifiers can be parameterized independently or shared across the $L$ layers, most early-exiting methods (Elbayad et al., 2020; Liu et al., 2021; Schuster et al., 2022) utilize the shared classifier due to its large number of parameters caused by enormous vocabulary size.

Table 1: Comparison of ROUGE-L scores between a full model, fine-tuned using all layer outputs, and *oracle*-exiting. We also measured cosine similarity between hidden states of the last layer and oracle-exited layer.

| Dataset | Model | Full M. | Oracle | Sim. |
|---|---|---|---|---|
| SAMSum | T5-small | 44.84 | 44.17 (-0.67) | 0.913 |
| | T5-large | 48.82 | 47.58 (-1.24) | 0.809 |
| CNN/DM | T5-small | 37.82 | 37.60 (-0.22) | 0.902 |
| | T5-large | 41.15 | 40.15 (-1.00) | 0.792 |
| Multi-News | LongT5-base | 37.62 | 29.63 (-7.99) | 0.724 |
| BIGPATENT | LongT5-base | 49.68 | 44.99 (-4.69) | 0.686 |

After the current token is early-exited at the $\ell^{\text{th}}$ layer, we need to calculate the key and value states for all deeper blocks in order to perform the self-attention for the subsequent tokens that pass through deeper blocks. For a more efficient approach of caching key and value states, the early-exiting frameworks employ the state copying mechanism. It duplicates the hidden states of the early-exited layer (*i.e.*, $\mathbf{h}_{t+1}^{i} = \mathbf{h}_{t+1}^{\ell}, \forall i \in [\ell+1, L]$), allowing us to compute the approximate key and value states required for the self-attention of Transformer networks. Schuster et al. (2022) have verified that state copying from lower layers does not have a detrimental effect on performance in the case of small-sized T5 models (Raffel et al., 2020).

## 4  Re-evaluating Early-exit Framework

In this section, we present *four* new findings from our re-evaluation of the early-existing framework. We utilized different model sizes of T5 (Raffel et al., 2020) on SAMSum (Gliwa et al., 2019) and CNN/DailyMail (See et al., 2017), and LongT5-base (Guo et al., 2022) architectures on Multi-News (Fabbri et al., 2019a) and BIG-PATENT (Sharma et al., 2019).

### 4.1  Lack of Robustness to Model Size and Output Sequence Length

We first re-evaluate the state copying mechanism which is an essential component of the early-exiting framework. Following Schuster et al. (2022), we use an *oracle* confidence measure that enables tokens to exit at the earliest layer, such that their predictions are identical to those of the final layer. Notably, as observed in Table 1, *the degradation of the generation quality with the state copying gets severe on larger models and datasets with the longer sequence* (▷ **Obs. 1**). For instance, when considering the oracle-exiting results, the T5-small model

demonstrates the degradation of only 0.67 on the SAMSum dataset, whereas the T5-large model experiences a much larger drop of 1.24. Similarly, on datasets such as Multi-News and BIGPATENT, which consist of relatively long output sequences, the oracle-exiting results exhibit a decrease of 7.99 and 4.69, respectively.

To strengthen the supporting evidence, we further discover the substantial variation in the distribution of hidden states across different layers. In Table 1, we also reported cosine similarity between the hidden states of the final layer and the oracle-exited layer. Even though the hidden states of the final and oracle-exited layers yield the same predictions, the cosine similarity between them decreases significantly as the decoder network gets larger and the output sequences become longer.

## 4.2 Performance Drop by Exit Position

To facilitate early-exiting for all decoder layers, the training objectives need to be a combination of the training objectives for each individual layer. We can present as follows:

$$\mathcal{L} = \sum_{i=1}^{L} \alpha_i \mathcal{L}_i \ \text{ where } \ \sum_i \alpha_i = 1, \qquad (1)$$

$\mathcal{L}_i$ and $\alpha_i$ is negative log-likelihood loss function and weight coefficient for $i^{\text{th}}$ layer, respectively. Especially, previous work set $\alpha_i$ as $1/L$ (unweighted average; Elbayad et al. 2020) or $i/\sum_i i$ (weighted average; Schuster et al. 2022). They demonstrated that these weighting rules effectively facilitate learning in earlier layers without compromising the overall performance of the full model on small-sized decoder models.

However, as shown in Figure 2, *we observed a notable decrease in the performance of static-exiting, which utilizes the same number of layers for all tokens, when utilizing only a small portion of the early layers from the T5-large model.* (▷ **Obs. 2**). For instance, if all tokens are exited in the first or second layers, the model achieved nearly zero ROUGE-L scores. Furthermore, when we apply the early-exiting framework to these models during inference, we verified that the T5-large model generates abnormally long sentences, actually consuming more inference time. Based on these results, in the subsequent experiments, we have excluded the first two or four layers from the candidates for early-exiting layers of base and large models, respectively.

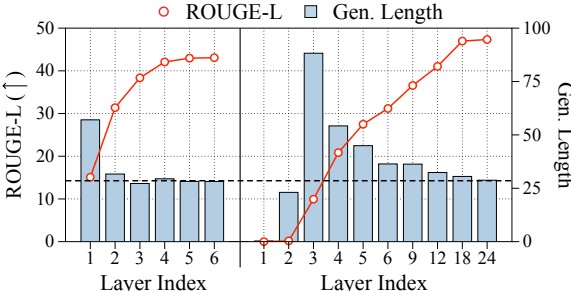

Figure 2: Illustration of the ROUGE-L scores and generated sequence length from the *static*-exiting approach in T5-small (left) and T5-large (right) on the SAMSum dataset. The horizontal dashed line represents the average sequence length of the ground truth.

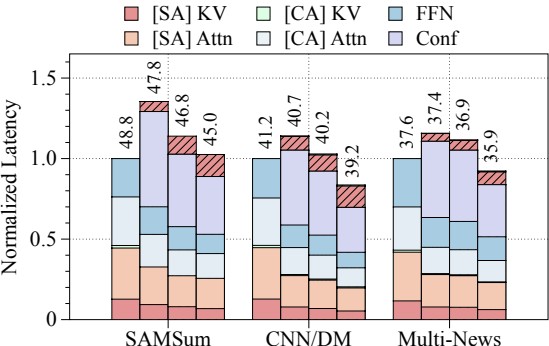

Figure 3: Component-wise computational cost on three datasets. Four bars correspond to full model and early-exiting with thresholds of 0.9, 0.7, and 0.5. The hatched color denotes the elapsed time after the token exits, related to the state copying mechanism. The numbers above the bars represent the ROUGE-L scores. SA and CA denote self- and cross-attention, respectively.

## 4.3 Non-negligible Computational Cost

During our analysis, we observed that the conventional early-exiting framework not only presents performance disadvantages but also poses challenges for inference latency. In Figure 3, we conducted a breakdown of the computational costs associated with a decoder model across three summarization datasets. Surprisingly, early-exiting has often shown an unexpected increase in total decoding time when compared to the baseline model without using early-exiting.

This can be attributed to *the non-negligible computational cost involved in measuring confidence at each layer*, particularly due to the softmax operations with the large vocabulary size. In addition, although the state copying method aims to reduce computation time in the MHA and FFN layers of the remaining layers, *the computation of key and value states using duplicated hidden states incurs additional non-negligible overhead* (▷ **Obs. 3**).

Table 2: The optimal confidence threshold to achieve desired performance. We chose the best values among threshold value from 0 to 1 with step size of 0.1. The numbers sequentially represent the selected threshold and corresponding performance (gray colored).

| Task | Dataset | Performance Drop | | |
|------|---------|-------|-------|--------|
| | | ~1% | ~5% | ~10% |
| SUM | SAMSum | 1.0 (48.8) | 0.7 (46.8) | 0.5 (45.0) |
| | CNN/DM | 1.0 (41.2) | 0.5 (39.2) | 0.3 (37.3) |
| | Multi-News | 0.8 (37.3) | 0.5 (35.9) | 0.4 (34.9) |
| | BIGPATENT | 1.0 (49.7) | 0.8 (47.3) | 0.6 (45.2) |
| QA | SQuAD | 0.1 (90.1) | 0.0 (88.3) | 0.0 (88.3) |
| MT | IWSLT | 1.0 (39.4) | 1.0 (39.4) | 1.0 (39.4) |

## 4.4 Disparate Optimal Confidence Threshold

Determining the appropriate threshold for exit confidence is a crucial challenge in the early-exiting framework as it directly impacts the trade-off between performance and latency (Zhang et al., 2019b; Schuster et al., 2022). As summarized in Table 2, our observations indicate that *the optimal confidence thresholds for achieving the lowest latency in the same performance significantly vary across datasets* (▷ **Obs. 4**). For instance, SQuAD and CNN/DailyMail datasets can maintain performance with relatively lower exit thresholds, whereas higher threshold values are required in the case of the IWSLT dataset. Previous work (Schuster et al., 2022) has leveraged distribution-free risk control techniques for confident generations. However, these methods require additional training time for statistical tests on the extra calibration set before the deployment, where time can be also influenced by the size of the threshold candidate sets.

## 5 Novel Early-Exiting Framework: FREE

Building upon the discoveries in Section 4, we introduce a Fast and Robust Early-Exiting framework named FREE, leveraging a shallow-deep module and capitalizing on the structure of parallel decoding. Furthermore, we present a confidence estimation algorithm designed to enhance the robustness of early-exiting within the FREE framework.

## 5.1 Shallow-Deep Module

We present an effective shallow-deep module, which strategically assigns a predetermined number of early layers ($L_S$) as a shallow model, while all the layers as a deep model. This module tackles the performance degradation associated with

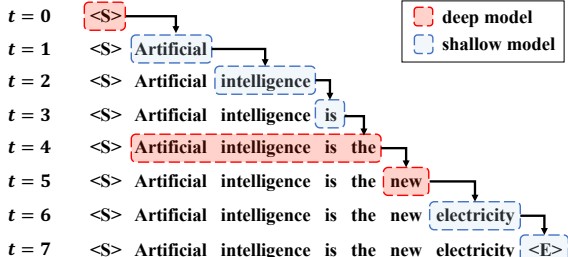

Figure 4: Overview of synchronized parallel decoding. We colored the tokens used to generate the next token based on the model that they forward.

co-training numerous exiting layers in the conventional early-exiting framework.

To enhance the performance of the shallow model, we exploit layerwise knowledge distillation (KD) as an additive loss term to Eq. (1) with $\alpha_{L_s} = L_s/(L + L_s)$ and $\alpha_L = L/(L + L_s)$:

$$\mathcal{L}_{KD} = \frac{1}{|L_S|} \sum_{i=1}^{L_S} \text{MSE}(\mathbf{H}_S^i, \mathbf{H}_D^{m(i)}),$$

where $m(i)$ indicates the layer in the deep model that extracts knowledge into the corresponding layer $i$ of the shallow model. $\mathbf{H}_S$ and $\mathbf{H}_D$ are hidden states from shallow and deep models.

We have experimented with the distillation from the last layer (KD-last; Wang et al. 2020; Ko et al. 2023), from fixed uniform mapped layers (KD-unif; Jiao et al. 2020; Park et al. 2021), and from dynamically mapped layers (KD-dyna; Xia et al. 2022). Especially, dynamic mapping function allows us to align each deep model layer with its closest counterpart in the shallow model:

$$m(i) = \arg\min_j \text{MSE}(\mathbf{H}_S^i, \mathbf{H}_D^j)$$

where $j$ denotes the layer indices of the deep model selected by the total number of $L_S$, and the condition of $m(1) \leq \cdots \leq m(L_S)$ should be satisfied. Based on the consistently superior performance of KD-dyna loss (see Appendix D.2), we utilized it for all experiments with the shallow-deep module.

## 5.2 Synchronized Parallel Decoding

We present synchronized parallel decoding as an alternative to the state copying mechanism, which is a key component of the conventional early-exiting framework but can lead to a significant performance decrease, as demonstrated in Section 4.1. In contrast to traditional approaches that have multiple exit points, our method incorporates the shallow-deep module, enabling us to stack consecutive

early-exited tokens in the shallow model until a non-exiting token is encountered. When decoding the token with the deep model, we enhance efficiency and effectiveness through parallel decoding, synchronously computing the key and value states of previously stacked tokens. The example of the parallel decoding process is depicted in Figure 4.

The underlying principle of this approach is to leverage the enhanced parallelism offered by modern hardware accelerators. This allows for efficient computations to be carried out simultaneously on the large number of sequences. Thus, by employing synchronized parallel decoding, we can directly compute multiple hidden states similar to a single token processing time. Besides, this can eliminate the potential performance degradation that may arise from inaccurate approximations of hidden states resulting from the state copying mechanism.

## 5.3 Adaptive Threshold Estimation

We propose a novel adaptive threshold estimation method that updates the threshold to be retailed for different datasets. Unlike the previous methods that utilize extra calibration sets (Schuster et al., 2022), we quickly adapt the threshold by using the information of early-stage instances, regardless of the initial threshold values. Especially, during parallel decoding, we collect samples to evaluate the correspondence between the confidence scores of the shallow model and the prediction alignment between shallow and deep models.

As depicted in Figure 1b, we observe that when the predictions of the deep and shallow models are identical, the confidence tends to skew towards one, otherwise it skews towards zero. To model this skewed distribution over $[0, 1]$, we utilize a *beta mixture* model (BMM; Ma and Leijon 2011) due to its flexibility and the appropriate support set of the beta distribution. The probability density function of beta distribution over $x \in [0, 1]$ is defined as:

$$p(x|\alpha, \beta) = \frac{\Gamma(\alpha + \beta)}{\Gamma(\alpha)\Gamma(\beta)} x^{\alpha-1}(1-x)^{\beta-1}$$

The parameters of the BMM are updated using the maximum likelihood estimator (MLE; Norden 1972) with observed data points.

$$\alpha_k = \bar{c}_k \left( \frac{\bar{c}_k(1-\bar{c}_k)}{s_k^2} - 1 \right), \beta_k = \frac{\alpha_k(1-\bar{c}_k)}{\bar{c}_k}, \quad (2)$$

where $\bar{c}_k$ being a average of the confidence $\{c_i^{L_s}\}_{i=1}^{|\mathcal{D}_c|}$ for corresponding $k$. $k$ is set to 1 if the predictions of the two models are identical, and 0

---

**Algorithm 1** Adaptive Threshold Estimation
**Input**: empty calibration dataset $\mathcal{D}_c$, initial confidence threshold $\lambda_c^0$, posterior condition $\zeta$, update number $T$
**Output**: updated confidence threshold $\lambda_c$
1: initialize $t \leftarrow 0$, $\lambda_c \leftarrow \lambda_c^0$
2: **while** $t \leq T$ **do**
3:     Generate $t^{\text{th}}$ sentence with $N_t$ tokens
4:     /* Update $\mathcal{D}_c$ */
5:     $\mathcal{D}_c \leftarrow \mathcal{D}_c \cup \{c_i^{L_s}, \mathrm{I}(\hat{y}_i^{L_s} = \hat{y}_i^L)\}_{i=1}^{N_t}$
6:     /* Find threshold with Eq.(2)-(4)*/
7:     $\alpha_k, \beta_k \leftarrow \mathrm{MLE_{BMM}}(\mathcal{D}_c)$ for $k \in \{0, 1\}$
8:     $\lambda_c \leftarrow \arg \min_{\lambda:p(k=1|\lambda) \geq \zeta} \lambda$
9:     update $t \leftarrow t + 1$
10: **end while**

---

otherwise. Similarly, $s_k$ is the standard deviation of confidence of related $k$.

$$\bar{c}_k = \frac{\sum_{i=1}^N \gamma_i c_i^{L_s}}{\sum_{i=1}^N \gamma_i}, \bar{s}_k^2 = \frac{\sum_{i=1}^N \gamma_i (c_i^{L_s} - \bar{c}_k)^2}{\sum_{i=1}^N \gamma_i}, \quad (3)$$

where $\gamma_i := \mathrm{I}(\hat{y}_i^{L_s} = \hat{y}_i^L)$ denote whether the prediction of two models are same.

After updating the BMM, we find an appropriate threshold for future tokens by identifying the point at which the posterior probability, defined as below, reaches $\zeta$:

$$p(k = 1|\lambda_c) = \frac{p(k=1)p(\lambda_c|\alpha_1, \beta_1)}{\sum_{j \in \{0,1\}} p(k=j)p(\lambda_c|\alpha_j, \beta_j)}. \quad (4)$$

Here, as we observe the severe imbalance between the case of $k = 0$ and 1, we restrict the prior value of each class to 0.5 for the balance between two cases (*i.e.*, $p(k = j) = 0.5 \ \forall j$). As this restriction makes us to use a smaller value of $\zeta$, we naïvely set it as 0.4. A detailed algorithm can be found in Algorithm 1.

## 6 Experiments

### 6.1 Experimental Setup

We conducted experiments on various sequence modeling tasks, including question answering (SQuAD; Rajpurkar et al. 2016b), machine translation (IWSLT 2017 En-De; Cettolo et al. 2017b), and text summarization tasks using SAMSum, CNN/DailyMail, Multi-News, and BIGPATENT datasets. The LongT5-base model was used for the Multi-News and BIGPATENT datasets, while the T5-large model was used for the other datasets. All implementations are based on PyTorch using Huggingface (Wolf et al., 2020; Lhoest et al., 2021). Further details can be found in Appendix B.

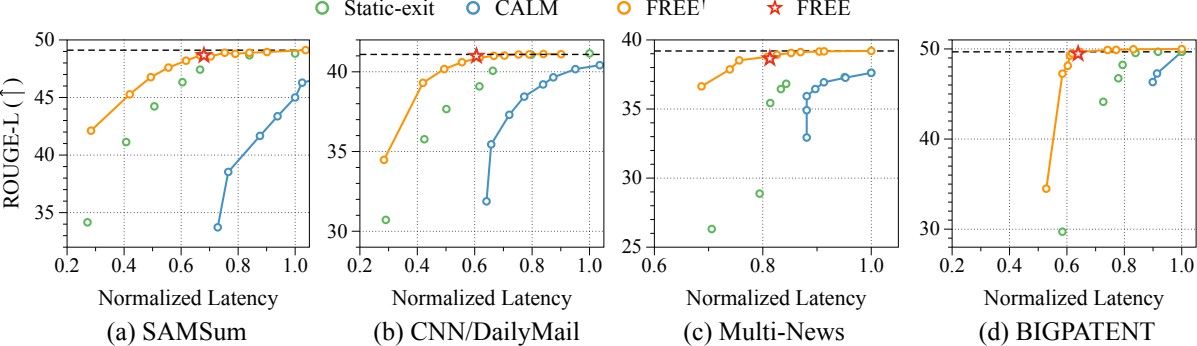

Figure 5: The trade-off between the generated output quality and normalized latency under different exit conditions. We varied the exit threshold values between 0 and 1 for both CALM and FREE[†] and the number of exit layers for the static-exiting framework. We exclude the inner point of the Pareto curve, and the dashed line represents the ROUGE-L score of the full model, which is the fine-tuned shallow-deep module.

Table 3: Comparison between early-exiting frameworks on various datasets. For CALM and FREE[†], we reported the performance using the smallest threshold value that achieves 99% performance of the full model, fine-tuned by weighted average or KD-dyna losses, respectively. The parentheses denote relative speedup based on the first row.

| Method | SUM | | | | QA | MT |
| --- | --- | --- | --- | --- | --- | --- |
| | SAMSum | CNN/DailyMail | Multi-News | BIGPATENT | SQuAD | IWSLT De-En |
| Full Model | 48.82 (×1.00) | 41.15 (×1.00) | 37.62 (×1.00) | 49.68 (×1.00) | 90.63 (×1.00) | 39.19 (×1.00) |
| CALM | 48.37 (×0.72) | 40.78 (×0.86) | 37.27 (×0.85) | 49.21 (×0.65) | 90.09 (×2.03) | 39.19 (×1.00) |
| Full Model | 49.11 (×1.00) | 41.09 (×1.00) | 39.20 (×1.00) | 49.68 (×1.00) | 91.90 (×1.00) | 39.39 (×1.00) |
| FREE[†] | 48.65 (×1.50) | 40.89 (×1.80) | 38.93 (×1.07) | 49.51 (×1.62) | 91.31 (×2.76) | 39.04 (×1.07) |
| FREE | 48.66 (×1.47) | 40.99 (×1.65) | 38.66 (×1.23) | 49.47 (×1.58) | 91.82 (×2.16) | 38.17 (×1.18) |

## 6.2 Experimental Results

In order to investigate the effect of the individual component of our proposed framework, we evaluate both FREE without and with an adaptive threshold estimator, denoted as FREE[†] and FREE.

**Overall performance.** In Figure 5, we present a comparison of the quality of generated output (ROUGE-L) and the inference latency between the FREE framework and baselines, including static-exiting and the conventional early-exiting method (CALM; Schuster et al. 2022). CALM method exhibited poorer performance compared to a simple static-exiting approach on all datasets, likely due to the state copying mechanism and the presence of numerous exit positions, as observed in Section 4. In contrast, FREE[†] demonstrated robust performance and the larger AUC (area under the curve) across datasets by adjusting exit thresholds.

**Adaptive threshold evaluation.** In the early-exiting framework, choosing the appropriate confidence threshold is crucial for achieving the best trade-off between generation quality and latency. Unlike previous calibration methods (Schuster et al., 2022) that require an extra calibration set

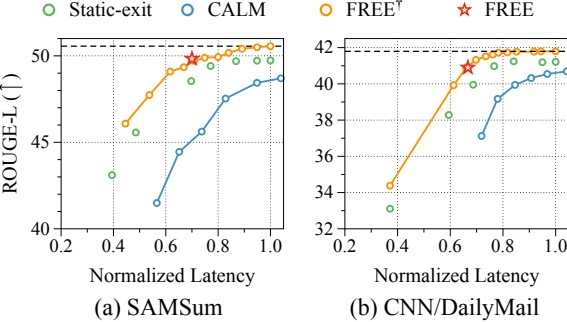

Figure 6: The trade-off between the generated output quality and normalized latency on T5-3B models.

and training time, our methodology effectively addresses this challenge by leveraging the byproduct of parallel decoding. As summarized in Table 3, FREE with adaptive threshold estimation successfully achieved significant speedup, by up to ×2.16, when preserving the 99% of full model performance. Furthermore, in Figure 5, the estimated threshold demonstrated nearly the maximum achievable speed improvement without sacrificing performance, represented by red stars.

**Large language models.** Recently, various studies (Dettmers et al., 2022; Xiao et al., 2023; Leviathan et al., 2023; Liu et al., 2023b) have

Table 4: The comparison of ROUGE-L and speedup based on different numbers of layers for shallow model and confidence thresholds.

| Dataset | $L_S$ | Threshold 0.7 | 0.5 | 0.3 |
|---|---|---|---|---|
| SAMSum | 4 | 48.27 (×1.04) | 46.95 (×1.09) | 44.72 (×1.15) |
| | 6 | 48.89 (×**1.32**) | 48.65 (×**1.50**) | 47.60 (×**1.80**) |
| | 8 | 48.74 (×1.11) | 47.97 (×1.17) | 47.09 (×1.31) |
| | 12 | **48.97** (×1.21) | **48.74** (×1.28) | **48.10** (×1.37) |
| CNN/DM | 4 | 41.03 (×1.45) | 40.59 (×1.68) | 39.88 (×1.86) |
| | 6 | 41.08 (×**1.53**) | 41.00 (×**1.69**) | 40.60 (×**2.07**) |
| | 8 | **41.19** (×1.44) | **41.15** (×1.64) | 40.95 (×1.69) |
| | 12 | 41.11 (×1.33) | 41.09 (×1.47) | **40.95** (×1.55) |

Table 5: The comparison between synchronized parallel decoding (SPC) and state copying (SC). The shallow-deep module is utilized in both decoding methods.

| Dataset | Method SC | SPD | Threshold 0.9 | 0.7 | 0.5 | 0.3 | 0.1 |
|---|---|---|---|---|---|---|---|
| SAMSum | ✓ | ✗ | 46.35 | 44.59 | 43.92 | 42.36 | 41.27 |
| | ✗ | ✓ | **48.89** | **48.89** | **48.65** | **47.60** | **45.27** |
| CNN/DM | ✓ | ✗ | 40.92 | 40.92 | 40.71 | 39.99 | 38.17 |
| | ✗ | ✓ | **41.12** | **41.08** | **41.00** | **40.60** | **39.30** |
| Multi-News | ✓ | ✗ | 38.43 | 37.61 | 36.55 | 33.99 | 29.34 |
| | ✗ | ✓ | **39.16** | **39.06** | **38.78** | **37.87** | **33.98** |

Table 6: The experimental results of FREE framework based on different sizes of the calibration set.

| Dataset | 3% Thr. | Perf. | Speed | 10% Thr. | Perf. | Speed | 100% Thr. |
|---|---|---|---|---|---|---|---|
| SAMSum | 0.51 | 48.66 | ×1.47 | 0.49 | 48.69 | ×1.51 | 0.48 |
| BIGPATENT | 0.54 | 49.47 | ×1.58 | 0.54 | 49.39 | ×1.63 | 0.54 |

aimed at boosting the inference speed of large language models (LLMs). To validate the applicability of the FREE framework on LLMs, we conducted experiments utilizing the T5-3B model (Raffel et al., 2020) on the SAMSum and CNN/DailyMail datasets. Due to substantial computational overhead, we utilized the LoRA adapter (Hu et al., 2022), targeting both self-attention and feed-forward layers with a rank of 64. Figure 6 summarized a comprehensive comparison of early-exiting methods. Our method maintained superiority over the baselines in terms of latency and ROUGE-L scores, showing the consistent performance trend observed in the T5-large model. Thus, we are confident that our proposed framework would demonstrate consistent level of inference acceleration, even with larger language models.

## 6.3 Ablation Study

**Different depth of shallow model.** In Table 4, we also ablate on the number of layers for the shallow model to observe the trade-offs. While our method demonstrated a trend towards higher speedup gains as the depth of the shallow model decreases, we experienced some decreases in performance and speed gain when the depth of the model is reduced too much (*e.g.,* four layers). We assumed that this is due to incorrect and redundant output sentences, similarly observed in the conventional early-exiting framework. Consequently, with enough depth (*e.g.,* six layers), FREE consistently showed robust performance and inference speedup.

**Robustness of parallel decoding.** In order to verify the robustness of our decoding mechanism, we conducted a comparative analysis between synchronized parallel decoding (SPD) and state copying (SC), both implemented with the shallow-deep

module. Synchronized parallel decoding consistently outperformed state copying across all three datasets by much higher ROUGE-L metrics, as summarized in Table 5. This improvement can be attributed to the updated hidden states that are obtained through the accurate computation of Transformer layers during parallel decoding. These findings suggest that our efficient decoding method for early-exited tokens can enhance the overall performance of the early-exiting framework as well.

**Dependency on size of calibration set.** By using the early-stage instances as the calibration set, we iteratively update the adaptive confidence threshold to converge to the appropriate value. Here, we have observed the sample efficiency of the adaptive threshold estimator by varying the sizes of this calibration set. Interestingly, even with only 3% of the total samples, our estimator can approximate the threshold, measured by the full sample set, as shown in Table 6. This ensures minimal additional computation time required for threshold estimation.

**Refining shallow model predictions.** Prior works (Leviathan et al., 2023; Chen et al., 2023; Kim et al., 2023) have proposed refinement methods to correct erroneous outputs from an approximation model. Specifically, when a wrong token is detected in previous sequences, they remove all subsequently generated tokens and restart the generation process from that point. In Table 7, we conducted experiments in order to evaluate the effects of this refinement method (Kim et al., 2023)

Table 7: The evaluation of refinement methods in the FREE framework. Refining thresholds control the level of acceptance for predictions from the shallow model.

| Dataset | Ref. | Thr. | Thr. 0.7 | | Thr. 0.3 | |
|---|---|---|---|---|---|---|
| | | | Perf. | Speed | Perf. | Speed |
| SAMSum | ✗ | - | 48.89 | ×**1.33** | 47.60 | ×**1.80** |
| | ✓ | 1.0 | **49.08** | ×1.26 | **48.31** | ×1.50 |
| | ✓ | 0.1 | 49.06 | ×1.17 | 48.27 | ×1.12 |
| CNN/DM | ✗ | - | **41.08** | ×**1.53** | 40.60 | ×**2.07** |
| | ✓ | 1.0 | 40.86 | ×1.51 | **40.78** | ×1.67 |
| | ✓ | 0.1 | 40.85 | ×1.35 | 40.75 | ×1.21 |

in our early-exiting framework. We observed that when the refinement threshold is set low, allowing for more correction by the deep model, the performance improvement is minimal compared to the increase in latency. Our findings suggest that these approaches that cannot guarantee an upper bound on latency increase may not be well-suited for integration into the early-exiting framework.

**Human-like summarization evaluation.** Recent studies (Gao et al., 2023; Liu et al., 2023a; Zhang et al., 2023) have argued that existing summarization evaluation metrics like ROUGE-L do not accurately represent the true summarization capabilities. Instead, they explored the human-like evaluation using LLMs based on their strong correlation with human judgment. Thereby, we conducted two human-like evaluation methods, Likert scale scoring and pairwise comparison (Gao et al., 2023), using ChatGPT API (gpt-3.5-turbo-0613). We compared a full model and our FREE framework on 100 instances, randomly drawn from the CNN/DailyMail dataset. Figure 7 and 8 provide the templates used for each evaluation task. For the full model, we observed scores of [4.73, 3.83, 3.87, 3.77], while our FREE method returned scores of [4.68, 3.84, 3.84, 3.72] across the four dimensions. Besides, the win counts for each method were 101 and 99, respectively. Given ROUGE-L scores of 41.09 (×1.00) for the full model and 40.99 (×1.65) for the FREE method, our method is certainly capable of yielding predictions of similar quality, while notably reducing computational overhead.

## 7 Conclusion

We proposed FREE framework to address the challenges of conventional early-exiting frameworks for autoregressive language models. Our approach incorporates three key components: (1) shallow-deep module, (2) synchronized parallel decoding,

Evaluate the quality of summaries written for a news article. Rate each summary on four dimensions: {Dimension_1}, {Dimension_2}, {Dimension_3}, and {Dimension_4}. You should rate on a scale from 1 (worst) to 5 (best).

Article: {Article}
Summary: {Summary}

Figure 7: The template for Likert scale scoring. The four dimensions are relevance, informativeness, fluency, and coherence.

Given a new article, which summary is better? Answer "Summary 0" or "Summary 1". You do not need to explain the reason.

Article: {Article}
Summary 0: {Summary_0}
Summary 1: {Summary_1}

Figure 8: The template for pairwise comparison. We measured twice by changing the order of summaries for a fair comparison.

and (3) adaptive threshold estimation. Through extensive experiments on various generation tasks, we empirically demonstrated the superior performance of FREE framework, achieving significant acceleration in latency without compromising the quality of the generated output.

**Limitations.** Our work addressed a fast and robust existing framework that can be efficiently utilized without concerns about performance degradation. However, our approach does have a few limitations which we discuss below: (1) Our method requires additional computational resources to fine-tune the shallow model. However, as we have demonstrated, parameter-efficient fine-tuning methods would be a promising solution to overcome this limitation. (2) While our work demonstrates robustness in the depth of the shallow model, further investigation is required to determine the appropriate depth for various language models. This aspect remains an area for additional research.

**Acknowledgement.** This work was supported by Institute of Information & communications Technology Planning & Evaluation (IITP) grant funded by Korea government (MSIT) [No. 2021-0-00907, Development of Adaptive and Lightweight Edge-Collaborative Analysis Technology for Enabling Proactively Immediate Response and Rapid Learning, 90%] and [No. 2019-0-00075, Artificial Intelligence Graduate School Program (KAIST), 10%].

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

## A  Dataset Description

We apply FREE on various generation tasks including summarization, question answering, and machine translation. We provide detailed descriptions of the datasets used.

- **SAMSum** (Summarization): SAMSum (Gliwa et al., 2019) consists of 16K messenger-like conversations that are annotated with a summary for providing a concise overview of the conversation's content in the third person.

- **CNN/DailyMail** (Summarization): CNN/ DailyMail (See et al., 2017) consists of over 300K English news articles that were originally designed for machine-reading and comprehension as well as abstractive question answering, but it now also supports extractive and abstractive summarization.

- **Multi-News** (Summarization): Multi-News (Fabbri et al., 2019a) comprises 45K news articles and corresponding summaries, where each summary is professionally crafted and provides links to the original articles referenced.

- **BIGPATENT** (Summarization): BIGPATENT (Sharma et al., 2019) contains 1.3M records of U.S. patent documents, each accompanied by abstractive summaries written by humans. In our work, we specifically focus on the Fixed Constructions category, which is one of the nine classification categories available in the dataset.

- **SQuAD** (Question Answering): The Stanford Question Answering (SQuAD, Rajpurkar et al. 2016b) is a collection of 87.6K reading comprehension tasks. It includes questions generated by crowd workers based on a set of Wikipedia articles.

- **IWSLT 2017** (Machine Translation): IWSLT 2017 (Cettolo et al., 2017b) addresses text translation, using a single machine translation (MT) system for multiple language directions such as English and German. Here, we specifically focus on a German-to-English translation task.

## B  Detailed Experimental Setup

**Training hyperparameters.**    In this section, we describe the detailed hyperparameter values for our work. We utilize the NVIDIA RTX 3090 GPUs for training the language models, and we summarize

Table 8:  Optimized hyperparameters for training shallow-deep T5 models. The column labeled '# Batch' indicates the product of the batch size per GPU and the number of GPUs. 'In len.' and 'Out len.' represent the maximum length of the input and output, respectively.

| Dataset | Model | # Batch | Epochs | In len. | Out len. |
|---|---|---|---|---|---|
| SAMSum | T5-large | 4×2 | 20 | 512 | 128 |
| CNN/DM | T5-large | 4×4 | 3 | 512 | 128 |
| Multi-News | LongT5-base | 2×2 | 3 | 2048 | 512 |
| BIGPATENT | LongT5-base | 2×2 | 3 | 2048 | 512 |
| SQuAD | T5-large | 4×2 | 10 | 512 | 30 |
| IWSLT 2017 | mT5-large | 4×4 | 2 | 1024 | 128 |

the training configuration in Table 8. For all dataset, we use AdaFactor (Shazeer and Stern, 2018) optimizer with the learning rate of 1e-4. For the adaptive threshold estimation, we set the initial threshold value $\lambda_c^0$ as 0.9, $\zeta$ as 0.4, $T$ as 3% of total sample number (refer to Algorithm 1).

**Performance metrics.**    To numerically measure the output quality of our method, we utilize the F1 score for SQuAD, BLEU score (Papineni et al., 2002) for IWSLT2017, and ROUGE score (Lin, 2004) for the four summarization tasks.

## C  Inference Latency Evaluation

For measuring inference speed, we execute 500 inference predictions for each dataset under each examined configuration in PyTorch (Paszke et al., 2019) compiled function in a single server with a single NVIDIA GeForce RTX 3039 GPU and 12th Gen Intel(R) Core(TM) i7-12700K CPU. For each inference prediction, we use batch size 1, which is a common use case for online serving (Schuster et al., 2022). Also, we use to generate output sequences through greedy sampling with a beam size of 1. We measure the time including all decoding steps until completion.

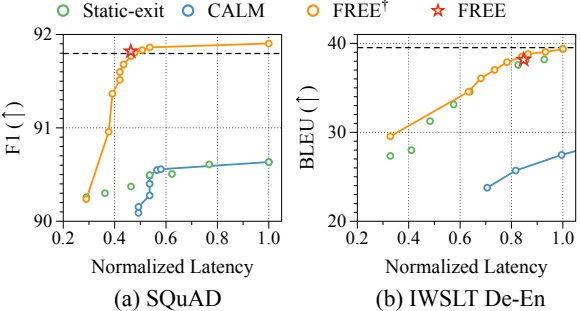

(a) SQuAD  (b) IWSLT De-En

Figure 9: The trade-off between the generated output quality and normalized latency under different exit conditions. The dashed line represents the F1 and BLEU scores of the full model, which is the fine-tuned shallow-deep module, respectively. Similar to Figure 5, we exclude the inner point of the Pareto curve.

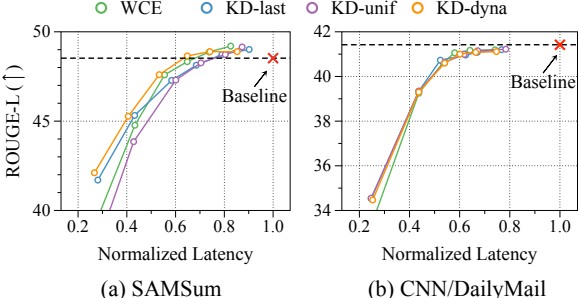

(a) SAMSum  (b) CNN/DailyMail

Figure 10: The trade-off between performance and normalized latency per sentence. We varied the exit thresholds in the range of {0.0, 0.1, 0.3, 0.5, 0.7, 0.9}. The latency values are normalized by the latency of a baseline, which is a simple fine-tuned full model.

## D  Additional Experimental Results

In this section, we provide additional experimental results to demonstrate the effectiveness of our proposed method and its individual components.

### D.1  Performance on Different Datasets

In this section, we present a comparison of the quality of the generated output (F1 or BLEU) and the inference latency on the SQuAD and IWSLT 2017 datasets, similar to the experiments in Figure 5. Figure 9 illustrates that both FREE† and FREE consistently outperform the CALM and static-exiting baselines in the SQuAD dataset, which aligns with our previous findings.

However, their performance advantages in the IWSLT dataset are slightly reduced compared to other datasets. This can be attributed to the larger vocabulary size of mT5 compared to T5, resulting in longer processing times for the confidence measurement. The CALM approach, which also utilizes large linear classifiers, exhibits much lower

Table 9: Comparison between FREE with T5-large and directly trained small-sized T5-base. We apply threshold values of FREE† as 0.1 for SQuAD and 0.2 for CNN/DailyMail.

| Method | Model | SQuAD | | CNN/DailyMail | |
|---|---|---|---|---|---|
| | | F1 | Speedup | ROUGE-L | Speedup |
| Full Model | T5-large | 91.82 | × 1.00 | 41.09 | × 1.00 |
| Full Model | T5-base | 90.50 | × 1.86 | 40.22 | × 2.06 |
| FREE† | T5-large | 90.95 | × 2.76 | 40.17 | × 2.07 |

effectiveness in this dataset as well. We believe that this challenge, regarding the large vocabulary size, can be mitigated by employing a vocabulary size-independent confidence measure that proposed in previous work (Schuster et al., 2022). Nonetheless, our proposed algorithm still outperforms the other baselines on various datasets.

### D.2  Layerwise Knowledge Distillation

Given the only two exit positions in our shallow-deep module, since their performance significantly impacts the overall robustness of the early-exiting approach, we carefully design the loss function for training. In Figure 10, we observed the performance trends of four different loss functions as we varied the exit thresholds. While the differences are not significant, the KD-dyna loss demonstrates better trade-offs compared to a weighted average or other KD-based losses. Specifically, the lower performance of KD-unif on the SAMSum dataset suggests that dynamically determining the layer mapping can facilitate more effective knowledge transfer between the deep and shallow models. Consequently, we trained our shallow-deep module using the KD-dyna loss for all experiments, and left the exploration of additional loss functions, such as contrastive distillation losses (Tian et al., 2019; Bae et al., 2021), for future work.

### D.3  Comparison with Small-sized Models

We conducted a comparison between the inference speed of FREE using T5-large model and a directly trained T5-base model. To ensure a fair comparison, we manually selected the appropriate confidence threshold for FREE† (without relying on an adaptive threshold estimator) to align its performance closely with that of T5-base. The results, presented in Table 9, demonstrate that our proposed method exhibited a competitive speedup in inference performance on the CNN/DailyMail dataset. Moreover, it demonstrated a superior F1 score and significantly higher speedup on the SQuAD dataset.

Table 10: Comparison between early-exiting frameworks on SAMSum with different decoding strategies.

| Method | top-k ($k = 50$) | | nucleus ($p = 0.92$) | |
|---|---|---|---|---|
| | ROUGE-L | Speedup | ROUGE-L | Speedup |
| Full Model | 44.34 | × 1.00 | 45.84 | × 1.00 |
| CALM | 42.35 | × 0.78 | 44.48 | × 0.82 |
| FREE | 43.58 | × 1.30 | 45.78 | × 1.31 |

We believe that the variance in speedup across the datasets can be attributed to the performance achievable by a directly trained smaller model, as well as a shallow model within the FREE framework. In the case of SQuAD, the T5-base model (12 layers) achieved a ROUGE-L score of 90.50, whereas a shallow model (6 layers) of our FREE framework yielded a similar score of 90.24. Our method effectively leverage these inherent benefits, thereby facilitating the inference speedup through exiting at lower layers.

## D.4 Various Decoding Strategies

To evaluate the applicability of FREE on various decoding methods, we conducted experiments with *top-k* sampling (Radford et al., 2019) and nucleus sampling (*top-p* sampling; Holtzman et al. 2020). *top-k* sampling samples the next word from the top $k$ most probable choices, instead of aiming to decode text that maximizes likelihood. On the other hand, nucleus sampling chooses from the smallest possible set of words whose cumulative probability exceeds the probability $p$. As detailed in Table 10, FREE method exhibited consistent and robust performance while achieving a larger speedup compared to CALM. These results affirm that our FREE framework can be widely applied, irrespective of the chosen decoding method.