# OpenReview forum: "Fast and Robust Early-Exiting Framework for Autoregressive Language Models with Synchronized Parallel Decoding"
_EMNLP/2023/Conference — EMNLP 2023 Main_

### Official Review · Reviewer_Bvg2 · 2023-08-02

**Soundness:** 4

**Excitement:**

4: Strong: This paper deepens the understanding of some phenomenon or lowers the barriers to an existing research direction.

**Missing References:**

- Some references that are missing:
    - “Deja Vu: Contextual Sparsity for Efficient LLMs at Inference Time” Liu et al.
    - “H2O: Heavy-Hitter Oracle for Efficient Generative Inference of Large Language Models” Zhang et al.
- You are using the library Datasets from Hugging Face but you are not mentioning it in the implementation details nor citing the corresponding paper “Datasets: A Community Library for Natural Language Processing” Lhoest et al.
- Please mind that several citations cite the ArXiv preprint and not the corresponding published article. Some examples are:
    - Leviathan et al. 2022 → ICML 2023
    - Elbayad et al. 2019 → ICLR 2020
    - Santilli et al. 2023 → ACL 2023
    - Nallapati et al. 2016 → CoNLL 2016

**Paper Topic And Main Contributions:**

The paper addresses the problem of speeding up the generation of autoregressive models via an early exit mechanism (FREE). Specifically, their approach addresses several limitations of previous early-exit works like (i) KV cache copying in exited layers, (ii) exit positioning, and (iii) tradeoff between latency and accuracy. Results show that FREE achieves a good tradeoff between latency and accuracy without requiring threshold selection.

**Questions For The Authors:**

- Since you are measuring the performance on GPU, I think you should call `torch.cuda.synchronize()` before calling the model in order to measure the wall clock time properly. I don’t see this implemented in the code, can you comment a little on this aspect? You should also use `time.perf_counter()` instead of `datetime.datetime.now()`.
- What kind of sampling procedure did you use for the different datasets? Are you using just greedy decoding or also top-p/top-k heuristics?
- In the paper, you said that you used a single 3090 GPU. However, in the readme file of your code repository you said “We support multi-GPUs training, while we utilized 1-4 GPUs”. Which one is correct?

**Reasons To Accept:**

- The paper is sound and well-written
- Experiments are throughout and exhaustive
- The method introduces contributions to improve the current limitations of early-exit approaches with several interesting observations
- The authors provided the code in a well-organized repository. Implementation details in the appendix are also well-documented and the used resources are reasonable (single Nvidia 3090 with desktop hardware)

**Reasons To Reject:**

The proposed approach FREE requires modification to the model and training

**Reproducibility:**

5: Could easily reproduce the results.

**Reviewer Confidence:**

4: Quite sure. I tried to check the important points carefully. It's unlikely, though conceivable, that I missed something that should affect my ratings.

**Typos Grammar Style And Presentation Improvements:**

Caption of Figure 1 can be improved. It does not explain the setting (a) and does not refer to (a) and (b).

---

> ### Author Rebuttal · Authors · 2023-08-28
>
> We would appreciate your insightful comments based on a thorough inspection of our paper. We are encouraged that you found our paper and code well-organized, experiments extensive and throughout, and the contributions important and interesting. The weakness and questions of our paper are discussed as follows.
>
> **W1. Requirement of modification to the model and training.**
>
> We would like to emphasize that there is no model modification because both shallow and deep decoders share one classifier. It only requires implementation details for exploiting synchronized parallel decoding.
>
> Besides, since shallow and deep models are jointly trained at once, it only leads to few additional training costs. Parameter-efficient fine-tuning methods could further help us to reduce the training costs. For example, we could achieve 49.99 and 39.50 ROUGE-L scores for deep and shallow models on SAMSum by utilizing the LoRA [1] (49.11 and 42.12 for full fine-tuning). Also, training a T5-xl model with LoRA showed 50.50 and 46.08, enabling FREE (with adaptive threshold estimator) to achieve 49.84 ROUGE-L score with ⨉1.50 latency speedup.
>
> **Q1. Question on implementation details when measuring latency.**
>
> Thanks to the feedback, we measured the inference speed on SAMSum again with  `torch.cuda.synchronize()` for more accurate measurement. The speedup trends were almost similar to previous results as follows (parentheses denote w/o synchronize).
>
>
> |             |    Layer 8    |    Layer 12   |    Layer 16   |    Layer 20   |    Layer 24   |
> |-------------|:-------------:|:-------------:|:-------------:|:-------------:|:-------------:|
> | Static-exit | ⨉2.97 (⨉3.00) | ⨉1.88 (⨉1.98) | ⨉1.45 (⨉1.50) | ⨉1.12 (⨉1.19) | ⨉1.00 (⨉1.00) |
> |             |   **Thres 0.1**   |   **Thres 0.3**   |   **Thres 0.5**   |   **Thres 0.7**   |   **Thres 0.9**   |
> | CALM        | ⨉1.26 (⨉1.37) | ⨉1.10 (⨉1.14) | ⨉1.02 (⨉1.00) | ⨉0.91 (⨉0.90) | ⨉0.82 (⨉0.77) |
> | FREE$^\dagger$       | ⨉2.35 (⨉2.38) | ⨉1.82 (⨉1.80) | ⨉1.47 (⨉1.50) | ⨉1.28 (⨉1.33) | ⨉1.17 (⨉1.19) |
>
>
>
> Moreover, we found that the difference between using `time.perf_counter()` and `datetime.datetime.now()` function is negligible.
>
> **Q2. Question on the results with top-p/top-k sampling.**
>
> We have used greedy decoding for all datasets due to its simplicity and effectiveness. When we measure the output performance with top-k and top-p sampling, our FREE (with adaptive threshold estimator) similarly shows robust performance and larger speedup compared to the CALM method. We summarized the results on the SAMSum dataset in the following table.
>
> |          |  |    top-k (k=50)     |       |  |   top-p (p=0.92)    |       |
> |---------|:---------------------:|:-------:|:-----:|:-----------------------:|:-----:|:-----:|
> |         |       Full      |   CALM  |  FREE |        Full       |  CALM |  FREE |
> | ROUGE-L |         44.34         | 42.35 | 43.58 |          45.84          | 44.48 | 45.78 |
> | Speedup |         ⨉1.00         |  ⨉0.78  | ⨉1.30 |          ⨉1.00          | ⨉0.82 | ⨉1.31 |
>
> **Q3. Question on computational resources.**
>
> We would note that we used a single 3090 GPU for the inference while we utilized 2 or 4 GPUs for fine-tuning the models.
>
> **Q4. Missing references and minor comments.**
>
> Thanks for the constructive comments. We will cite additional related publications, and modify some citation formats and the caption of Figure 1 in the finalized version.
>
> **References**
>
> [1] Hu, Edward J., et al. "LoRA: Low-Rank Adaptation of Large Language Models." International Conference on Learning Representations (2021).

---

### Official Review · Reviewer_ZsSB · 2023-08-04

**Soundness:** 5

**Excitement:**

4: Strong: This paper deepens the understanding of some phenomenon or lowers the barriers to an existing research direction.

**Paper Topic And Main Contributions:**

The authors propose a new mechanism for early-exiting during decoding at inference time in Autoregressive language models. Their framework utilizes a shallow-deep division of the LM which also enables synchronized parallel decoding, as well as on-the-fly threshold estimator that ensures that low-confidence early exits don't seep through.

**Reasons To Accept:**

1. Good overview of current early exiting frameworks, their strengths and weaknesses in Section 4.
2. Novel early exiting method that tackles the biggest problems with existing frameworks, including low quality output using an adaptive threshold.

**Reasons To Reject:**

1. The authors only present ROUGE-L results over time for the various models. However, I believe at least a  qualitative human evaluation is necessary when evaluating generation based tasks/models. This would ensure that good ROUGE-L scores are not a result of idiosyncrasies in the data, output or the ROUGE-L metric itself. Moreover, it would be interesting to see! Do the authors mean to say that the outputs are completely similar, or of similar quality?

**Reproducibility:**

4: Could mostly reproduce the results, but there may be some variation because of sample variance or minor variations in their interpretation of the protocol or method.

**Reviewer Confidence:**

3: Pretty sure, but there's a chance I missed something. Although I have a good feel for this area in general, I did not carefully check the paper's details, e.g., the math, experimental design, or novelty.

---

> ### Author Rebuttal · Authors · 2023-08-28
>
> We are grateful that you have acknowledged our extensive analysis on the current early-exiting framework and novelty of the proposed FREE method. The weakness of our paper is discussed as follows.
>
> **W1. Qualitative human evaluation results.**
>
> As recent works showed the effectiveness of  LLM evaluation and high correlation to human evaluation [1-4],  we conducted two human-like summarization evaluations with ChatGPT API (gpt-3.5-turbo-0613), Likert scale scoring and pairwise comparison [2]. Here, we compared a full model and our FREE framework (with adaptive threshold estimation) on 100 randomly selected instances from CNN/DM. Note that they showed 41.09 (⨉1.00) and 40.99 (⨉1.65) ROUGE-L scores, respectively. The detailed evaluation setup and results are described as follows.
>
> (1) For measuring the ***Likert scale scoring***, we used the following template:
>
> *“Evaluate the quality of summaries written for a news article. Rate each summary on four dimensions: relevance, informativeness, fluency, and coherence. You should rate on a scale from 1 (worst) to 5 (best).”*
>
> We obtained [4.73, 3.83, 3.87, 3.77] for a full model and [4.68, 3.84, 3.84, 3.72] for our FREE in four dimensions, respectively.
>
> (2) For measuring the ***Pairwise comparison***, we used the following template:
>
> *“Given a new article, which summary is better? Answer “Summary 0” or “Summary 1”. You do not need to explain the reason.”*
>
> We got a win count of 101 and 99 for the full model and FREE, measured twice by changing the order of summaries for the fair comparison.
>
> As a result, we are confident that our method is capable of yielding predictions of similar quality with those from a full model, while reducing computational overhead.
>
> **References**
>
> [1] Liu, Yang, et al. "Gpteval: Nlg evaluation using gpt-4 with better human alignment." arXiv preprint arXiv:2303.16634 (2023).
> [2] Gao, Mingqi, et al. "Human-like summarization evaluation with chatgpt." arXiv preprint arXiv:2304.02554 (2023).
> [3] Min, Sewon, et al. "FActScore: Fine-grained Atomic Evaluation of Factual Precision in Long Form Text Generation." arXiv preprint arXiv:2305.14251 (2023).
> [4] Zhang, Xinghua, et al. "Wider and Deeper LLM Networks are Fairer LLM Evaluators." arXiv preprint arXiv:2308.01862 (2023).

---

### Official Review · Reviewer_Tuxi · 2023-08-09

**Typos Grammar Style And Presentation Improvements:** n/a
**Soundness:** 3

**Excitement:**

4: Strong: This paper deepens the understanding of some phenomenon or lowers the barriers to an existing research direction.

**Missing References:**

n/a

**Paper Topic And Main Contributions:**

This paper proposes an early-exiting framework for autoregressive language models, which allows for early exits based on confidence. To cache key and value states for early-exit tokens, the authors copy the lower layer hidden states into higher layers, following Schuster et al. (2022). The proposed method increases the inference speed in QA, summarization, and MT with minimal performance losses.

**Questions For The Authors:**

n/a

**Reasons To Accept:**

The proposed method effectively accelerates latency, particularly in QA tasks.

**Reasons To Reject:**

It would be beneficial if the authors could include a baseline of directly training a smaller model (with similar average inference speed to the early-exit framework). Such a model may achieve similar performance compared to the proposed method.

**Reproducibility:**

4: Could mostly reproduce the results, but there may be some variation because of sample variance or minor variations in their interpretation of the protocol or method.

**Reviewer Confidence:**

4: Quite sure. I tried to check the important points carefully. It's unlikely, though conceivable, that I missed something that should affect my ratings.

---

> ### Author Rebuttal · Authors · 2023-08-28
>
> We thank you for your constructive comments, and we are encouraged that you pointed out the effectiveness of our proposed method on accelerating inference latency. The weakness of our paper is discussed as follows.
>
> **W1. Comparison to a baseline of directly training a smaller model.**
>
> We compared our FREE results utilizing T5-large with those of directly trained T5-base (full model). Since the T5-base model has lower performance and latency compared to T5-large, we selected the proper confidence threshold of FREE$^\dagger$ to match the similar performance (*e.g.*, 0.1 for SQuAD and 0.2 for CNN/DM). The experimental results are as follows.
>
> |                 | SQuAD |         | CNN/DailyMail |         |
> |-----------------|:-----:|:-------:|:-------------:|:-------:|
> |                 |   F1  | Speedup |    ROUGE-L    | Speedup |
> | Full (T5-large) | 91.82 |  ⨉1.00  |     41.09     |  ⨉1.00  |
> | Full (T5-base)  | 90.50 |  ⨉1.86  |     40.22     |  ⨉2.06  |
> | FREE$^\dagger$    | 90.95 |  ⨉2.76  |     40.17     |  ⨉2.07  |
>
> Our proposed method demonstrated competitive inference speedup on the CNN/DM summarization task, and showed superior F1 scores with much higher speedup on the SQuAD task.
>
> We would note that FREE also can be used with smaller models, and the static-exit baselines are similar to smaller model cases because they deterministically use a small number of decoder layers of original T5-large (refer to Figure 5).

---

### Official Review · Reviewer_e6CC · 2023-08-13

**Soundness:** 3

**Excitement:**

3: Ambivalent: It has merits (e.g., it reports state-of-the-art results, the idea is nice), but there are key weaknesses (e.g., it describes incremental work), and it can significantly benefit from another round of revision. However, I won't object to accepting it if my co-reviewers champion it.

**Missing References:**

[1] Ghazvininejad, Marjan et al. “Mask-Predict: Parallel Decoding of Conditional Masked Language Models.” Conference on Empirical Methods in Natural Language Processing (2019).

[2] Cho, Kyunghyun. “Noisy Parallel Approximate Decoding for Conditional Recurrent Language Model.” ArXiv abs/1605.03835 (2016): n. pag.

[3] Guo, Junliang et al. “Incorporating BERT into Parallel Sequence Decoding with Adapters.” ArXiv abs/2010.06138 (2020): n. pag.

[4] Gu, Jiatao et al. “Trainable Greedy Decoding for Neural Machine Translation.” Conference on Empirical Methods in Natural Language Processing (2017).

[5] Angela Fan, Edouard Grave, and Armand Joulin. Reducing transformer depth on demand with
structured dropout. arXiv preprint arXiv:1909.11556, 2019.

**Paper Topic And Main Contributions:**

The authors proposed a Fast and Robust Early-Exiting (FREE) framework that incorporates a shallow-deep module and synchronized parallel decoding. The framework allows for faster inference by synchronizing the decoding process and introduces a novel adaptive threshold estimator. Experimental results demonstrate the effectiveness of the framework in summarization and machine translation tasks.

**Reasons To Accept:**

1. The paper is well organized.
2. Experiments are solid and comprehensive.

**Reasons To Reject:**

1. Missing references. Please refer to the "Missing References" section.
2. Contributions are limited since the parallel decoding problem is well-studied in the literature. There is a lack of comparisons between the proposed framework and other decoding techniques.

**Reproducibility:**

4: Could mostly reproduce the results, but there may be some variation because of sample variance or minor variations in their interpretation of the protocol or method.

**Reviewer Confidence:**

4: Quite sure. I tried to check the important points carefully. It's unlikely, though conceivable, that I missed something that should affect my ratings.

---

> ### Author Rebuttal · Authors · 2023-08-28
>
> We thank you for your thoughtful feedback. We are encouraged that you found our paper well-organized, and our experimental results comprehensive. The weaknesses of our paper are discussed as follows.
>
>
> **W1. Some missing references.**
>
> We would appreciate it for informing relevant previous works, related to novel decoding methods with less computational overhead. We will add these references in our paper.
>
> **W2. Contributions of paper and comparisons to other decoding techniques.**
>
> The contributions of our paper lie in analyzing the challenges of conventional early-exiting frameworks within autoregressive language models. These challenges are effectively mitigated through introducing parallel decoding or the adaptive threshold estimator. While we are motivated from existing parallel decoding techniques, our work has tailored parallel decoding to suit the early-exiting framework, switching autoregressive decoding to parallel decoding using stacked representations.
>
> We also discussed the efficacy and superiority of our method in Table 7 compared to speculative decoding [1,2], a recent and powerful decoding technique aimed at enhancing efficiency. Moreover, we would like to emphasize that we have compared our FREE framework with existing methods [3,4].
>
>
> **References**
>
> [1] Leviathan, et al. "Fast inference from transformers via speculative decoding." International Conference on Machine Learning (2023).
> [2] Kim, Sehoon, et al. "Big little transformer decoder." arXiv preprint arXiv:2302.07863 (2023).
> [3] Elbayad, Maha, et al. "Depth-adaptive transformer." Eighth International Conference on Learning Representations (2020).
> [4] Schuster, Tal, et al. "Confident adaptive language modeling." Advances in Neural Information Processing Systems 35 (2022).

---

### Meta-Review · Area_Chair_MHyh · 2023-09-17

**Recommendation:** 5

**Metareview:**

The paper works on Early-exit based strategy to accelerate the model. Three main improvements: 1) shallow-deep module 2) synchronized parallel decoding 3) threshold estimation without valid dataset.

Pros:

1. Well organized and extensive experiments to validate the methods.

2. Provide a very good overview over Early-exit based methods.

3. Observations on the existing early-exit methods are insightful. It could potentially benefit others to further improve this line of reserach.

4. The threshold estimation is novel.

Cons:

Section 5.2 is not well written. There is no math formula or enough English explanation on how the parallel decoding works. Figure 4 itself is not enough for readers to quickly understand.

---

### Decision · Program_Chairs · 2023-10-07

**Decision:**

Accept-Main

**Comment:**

The paper works on Early-exit based strategy to accelerate the model. Three main improvements: 1) shallow-deep module 2) synchronized parallel decoding 3) threshold estimation without valid dataset.

Pros:

1. Well organized and extensive experiments to validate the methods.

2. Provide a very good overview over Early-exit based methods.

3. Observations on the existing early-exit methods are insightful. It could potentially benefit others to further improve this line of reserach.

4. The threshold estimation is novel.

Cons:

Section 5.2 is not well written. There is no math formula or enough English explanation on how the parallel decoding works. Figure 4 itself is not enough for readers to quickly understand.